# Self-Supervised CSF Inpainting with Synthetic Atrophy for Improved Accuracy Validation of Cortical Surface Analyses

**Jiacheng Wang**[*1]                                          JIACHENG.WANG.1@VANDERBILT.EDU
**Kathleen E. Larson**[*2]                                  KATHLEEN.E.LARSON@VANDERBILT.EDU
**Ipek Oguz**[1]                                                      IPEK.OGUZ@VANDERBILT.EDU
[1] *Department of Computer Science, Vanderbilt University, Nashville, USA*
[2] *Department of Biomedical Engineering, Vanderbilt University, Nashville, USA*

**Editors:** Accepted for publication at MIDL 2023

## Abstract

Accuracy validation of cortical thickness measurement is a difficult problem due to the lack of ground truth data. To address this need, many methods have been developed to synthetically induce gray matter (GM) atrophy in an MRI via deformable registration, creating a set of images with known changes in cortical thickness. However, these methods often cause blurring in atrophied regions, and cannot simulate realistic atrophy within deep sulci where cerebrospinal fluid (CSF) is obscured or absent. In this paper, we present a solution using a self-supervised inpainting model to generate CSF in these regions and create images with more plausible GM/CSF boundaries. Specifically, we introduce a novel, 3D GAN model that incorporates patch-based dropout training, edge map priors, and sinusoidal positional encoding, all of which are established methods previously limited to 2D domains. We show that our framework significantly improves the quality of the resulting synthetic images and is adaptable to unseen data with fine-tuning. We also demonstrate that our resulting dataset can be employed for accuracy validation of cortical segmentation and thickness measurement. Our code is publicly availiable at https://github.com/MedICL-VU/SSL-CSF-Inpainting

**Keywords:** Inpainting, Self-supervised, Sinusoidal positional encoding, Edge detection, Synthetic atrophy, Multi-modal MRI, Cortical thickness measurement

## 1. Introduction

Cortical segmentation and thickness measurement are powerful tools for measuring changes in the gray matter (GM) of the brain (Fischl and Dale, 2000; Han et al., 2004; Oguz and Sonka, 2014b,a; Reuter et al., 2012). Evaluating the accuracy of these methods is difficult due to the lack of ground truth data. Many have proposed synthetic atrophy methods as a solution to generate longitudinal data with known changes in cortical thickness (CT) (Bernal et al., 2021; Davatzikos et al., 2001; Freeborough and Fox, 1997; Karaçali and Davatzikos, 2006; Khanal et al., 2017; Xia et al., 2019; Larson and Oguz, 2022, 2021). Several of these techniques (Davatzikos et al., 2001; Khanal et al., 2017; Larson and Oguz, 2022, 2021; Karaçali and Davatzikos, 2006) operate using a deformable registration to move the outer GM boundary in towards the white matter (WM), effectively shrinking the GM while expanding the cerebrospinal fluid (CSF). However, these registration-based methods often induce GM/CSF boundary blurring in the atrophied images due to interpolation.

---

[*] Contributed equally

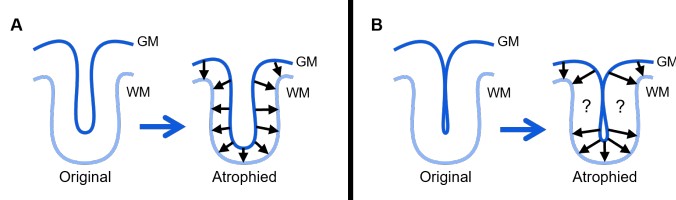

Figure 1: Motivation for CSF inpainting in registration-based synthetic atrophy (RBSA). **(A)** RBSA works as intended when the CSF inside a sulcus is visible. **(B)** RBSA fails in narrow sulci where CSF is difficult to resolve due to partial volume effects.

These methods can also struggle within deep sulci where CSF is not visible pre-atrophy due to partial volume effects, which prevents deformable models from generating CSF in the atrophied result despite GM thinning (Fig. 1). To overcome these problems, we propose the use of inpainting with a self-supervised deep-learning model to generate CSF in these atrophied regions and create images with more plausible GM/CSF boundaries.

Inpainting is often used to replace missing or corrupted image regions. While traditional methods (Barnes et al., 2009; Li et al., 2017; Fleishman et al., 2018) can produce high-quality images, they are limited in their ability to semantically reconstruct the missing parts of an image (Zhou et al., 2021). Deep learning-based methods (Pathak et al., 2016; Iizuka et al., 2017; Nazeri et al., 2019; Zhang et al., 2020; Song et al., 2018; Xu et al., 2021; Ren et al., 2019; Suvorov et al., 2022) have shown improved results, especially in larger missing regions. However, these are primarily restricted to 2D domains, rendering them unable to maintain consistency across image slices when applied to 3D volumes. Many of these methods also require extensive training data, which can be prohibitive in new datasets.

In this paper, we present a 3D GAN model for CSF inpainting. First, we learn to synthesize MRIs given the desired tissue labels using a patch-based dropout scheme in real, i.e., not atrophied, images. This allows us to obtain ground truth to supervise our training. Additionally, since the dropout masks contain not just CSF but all tissue types, the model learns to synthesize tissue boundaries realistically. Since our downstream goal is cortical surface reconstruction, a plausible GM/CSF boundary is critical. Once trained, we apply our model to synthetically atrophied images obtained using a registration-based method. Our main contributions are:

1. The use of CSF inpainting for synthetic cortical atrophy induction is the first of its kind, and provides an interesting new challenge for the MIDL community.

2. Our model can handle multi-label inpainting: we extend a previous method (Pathak et al., 2016) to include tissue labels within dropout regions rather than a binary mask.

3. We combine several existing 2D methods and extend them to 3D, including a self-supervised dropout scheme (Pathak et al., 2016), edge priors (Nazeri et al., 2019), and sinusoidal positional encoding (SPE) (Xu et al., 2021).

4. Our model synthesizes better images than 2D methods and can adapt to unseen data.

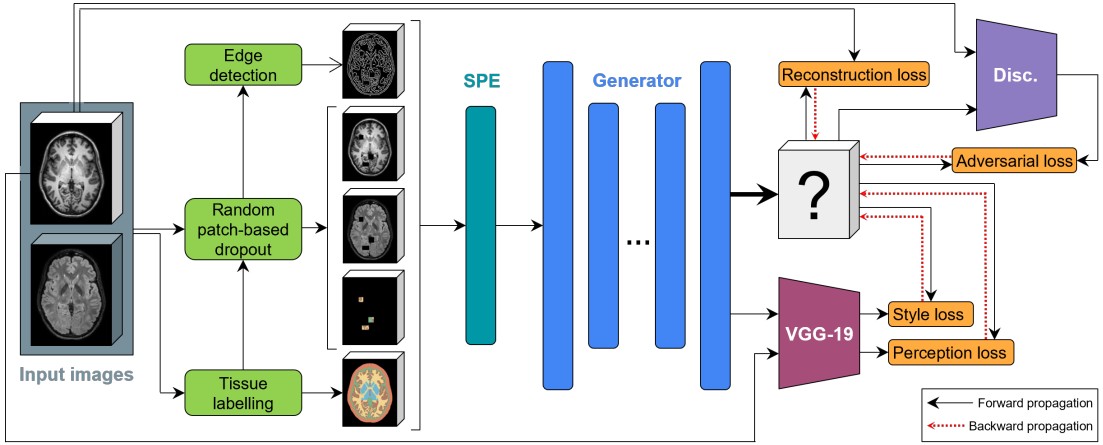

Figure 2: Self-supervised training pipeline for a T1w. We pre-processed (green) T1w and FLAIR images (gray) to obtain 5 input channels. We used 3D SPE and adversarial training with a 3D generator and discriminator. Loss functions (orange) were computed using a VGG-19 network, supervised by the original T1w.

## 2. Methods

Fig. 2 describes our pipeline. We derived 5 input channels per training subject (Sec. 2.2), and used a 3D GAN model (Sec. 2.3) with adapted 3D SPE. We performed cross-validation training and fine-tuning for unseen data (Sec. 2.4) with public datasets (Sec. 2.1).

### 2.1. Data

**NITRC Kirby Dataset (Kirby).** The Kirby dataset (n=21) (Landman et al., 2011) is a test-retest dataset of healthy controls. Each subject is associated with two sets of MPRAGE/T1w ($1.0 \times 1.0 \times 1.2$ mm$^3$) and FLAIR ($1.1 \times 1.1 \times 1.1$ mm$^3$) images procured on the same day; each set was used as an independent training sample.
**Validation Data for Cortical Reconstruction Algorithms (VDCRA).** The VDCRA (Shiee et al., 2014) (n=10) consists of 5 healthy controls (HC) and 5 multiple sclerosis (MS) subjects, each with a T1w and a FLAIR image, with the same resolutions as the Kirby dataset. Additionally, each subject is associated with landmarks denoting the WM and GM cortical surfaces in seven different regions on either hemisphere (1,680 per subject).

### 2.2. Generation of Input Image Channels

We trained two separate models to inpaint T1w and FLAIR images, respectively. Each model took five input channels (Fig. 3), with $H \times W \times D$ voxels each, as described below.
**MRI preprocessing.** The T1w and FLAIR images, denoted $\mathbf{I}_{T1}$ and $\mathbf{I}_{FL}$, were pre-processed using the FreeSurfer program (FS) (Dale et al., 1999) to perform isotropic resampling, intensity normalization, and bias field correction. Although we trained a separate model for each modality, both models used both $\mathbf{I}_{T1}$ and $\mathbf{I}_{FL}$ as inputs.

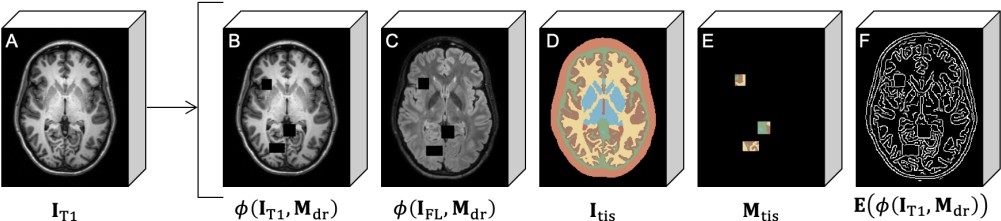

$$\mathbf{I}_{T1} \qquad \phi(\mathbf{I}_{T1}, \mathbf{M}_{dr}) \qquad \phi(\mathbf{I}_{FL}, \mathbf{M}_{dr}) \qquad \mathbf{I}_{tis} \qquad \mathbf{M}_{tis} \qquad \mathbf{E}(\phi(\mathbf{I}_{T1}, \mathbf{M}_{dr}))$$

Figure 3: Example input channels for a target T1w image.

**Random patch-based dropout masks.** We generated an image $\mathbf{M}_{dr}$ with a random number of randomly placed dropout patches. These patches were constrained such that their combined volume was less than 1% of the total image dimension (Tang et al., 2022). Patches containing background voxels were rejected. We then defined a function $\phi(\cdot)$ that replaced dropout voxels with Gaussian noise $\mathcal{N}(0,1)$. Fig. 3B/C depicts the dropout images $\phi(\mathbf{I}_{T1}, \mathbf{M}_{dr})$ and $\phi(\mathbf{I}_{FL}, \mathbf{M}_{dr})$, respectively. Further details can be found in Apdx. B.1.
**Whole-brain tissue labels.** We used FS to compute a skull-strip mask and a tissue label map (aseg.mgz) for each training sample. These were converted into a custom labelling, denoted $\mathbf{I}_{tis}$, with the following classes: cortical GM, WM, non-ventricle CSF, ventricles, non-cortical brain structures, head (remaining foreground voxels outside the skull-strip mask), and background (Fig. 3D). We also created an auxiliary label map $\mathbf{M}_{tis}$ that retains only the tissue labels within the dropout patches, i.e., $\mathbf{M}_{tis} = \mathbf{I}_{tis} * \mathbf{M}_{dr}$ (Fig. 3E).
**Edge detection.** The last input channel is the edge map of the target image (i.e., $\phi(\mathbf{I}_*, \mathbf{M}_{dr})$), where $\mathbf{I}_*$ refers to $\mathbf{I}_{T1}$ or $\mathbf{I}_{FL}$). A 2D Canny edge detector (Canny, 1986) was applied to each axial slice; these edge slices were stacked into a 3D volume $\mathbf{E}(\phi(\mathbf{I}_*, \mathbf{M}_{dr}))$ (Fig. 3F).

### 2.3. 3D GAN Model

**Network architecture.** Our GAN network possessed a typical encoder-decoder structure, but with 3D convolutions rather than 2D. It consisted of a generator $\mathbf{G}$ and a discriminator $\mathbf{D}$. The generator $\mathbf{G}$ (Apdx. B.2.1) used the five input channels to synthesize the entire image $\mathbf{I}_{pred,*}$ at each iteration of adversarial training. We discarded the synthesized voxels outside the dropout mask and replaced them by their original value in the input image. The discriminator (Apdx. B.2.2) then classified this composite output as "real" or "fake".
**Sinusoidal Positional Encoding.** Recent research (Xu et al., 2021) demonstrated that the use of explicit positional encoding in generative models is crucial for maintaining a spatial inductive bias balanced across the entire image and consistent transformations between positions. We extended SPE to 3D with the following structure:

$$\underbrace{[sin(\omega_0 i), cos(\omega_0 i), \ldots]}_{\text{sagittal direction}}, \underbrace{[sin(\omega_0 j), cos(\omega_0 j), \ldots]}_{\text{coronal direction}}, \underbrace{[sin(\omega_0 k), cos(\omega_0 k), \ldots]}_{\text{axial direction}} \qquad (1)$$

Here, $\omega_0 = 1/10000^{6/N}, i, j, k \in \mathbb{Z}$ in $[0, \lceil N/6 \rceil]$, for $N$ input channels ($N = 5$ in our case).
**Loss Function.** Inspired by previous work (Nazeri et al., 2019), our model was trained using a loss function $\mathcal{L}_{train} = \lambda_1 \mathcal{L}_{recon} + \lambda_2 \mathcal{L}_{adv} + \lambda_3 \mathcal{L}_{\bar{perc}} + \lambda_4 \mathcal{L}_{\bar{sty}}$.

Reconstruction loss $\mathcal{L}_{\mathrm{recon}}$ computes the dissimilarity between the original $\mathbf{I}_*$ and the reconstructed $\mathbf{I}_{\mathrm{pred},*}$. Adversarial loss $\mathcal{L}_{\mathrm{adv}}$ encourages $\mathbf{G}$ to generate more realistic images. Perceptual loss $\mathcal{L}_{\mathrm{p\bar{e}rc}}$ and style loss $\mathcal{L}_{\mathrm{s\bar{t}y}}$ were both calculated using a 2D VGG-19 network pre-trained on ImageNet (Simonyan and Zisserman, 2014), and are computed within axial slices $s \in S^{\mathrm{axial}}$ rather than in 3D. $\mathcal{L}_{\mathrm{p\bar{e}rc}}$ seeks to improve perceptual similarity of $\mathbf{I}_*$ and $\mathbf{I}_{\mathrm{pred},*}$ by penalizing differences between their VGG features. $\mathcal{L}_{\mathrm{s\bar{t}y}}$ aims to make the VGG features that activate together similar between the two images (Johnson et al., 2016).

$$\mathcal{L}_{\mathrm{recon}} = \mathbb{E}_{\mathbf{M}_{\mathrm{dr}}} ||\mathbf{I}_* - \mathbf{I}_{\mathrm{pred},*}||_1 \tag{2}$$

$$\mathcal{L}_{\mathrm{adv}} = \underbrace{\mathbb{E}_{(\mathbf{I}_*,\mathbf{I}_{\mathrm{tis}})} [\log \mathbf{D}(\mathbf{I}_*, \mathbf{I}_{\mathrm{tis}})]}_{\text{is the original image real or fake?}} + \underbrace{\mathbb{E}_{(\mathbf{I}_{\mathrm{pred},*},\mathbf{I}_{\mathrm{tis}})} [\log(1 - \mathbf{D}(\mathbf{I}_{\mathrm{pred},*}, \mathbf{I}_{\mathrm{tis}}))]}_{\text{is the reconstructed image real or fake?}} \tag{3}$$

$$\mathcal{L}_{\mathrm{p\bar{e}rc}} = \mathbb{E}_{s \in S^{\mathrm{axial}}} \underbrace{\left[ \mathbb{E}_i \left[ ||\psi_i(\mathbf{I}_*^s) - \psi_i(\mathbf{I}_{\mathrm{pred},*}^s)||_1 \right] \right]}_{\text{VGG feature similarity}} \tag{4}$$

$$\mathcal{L}_{\mathrm{s\bar{t}y}} = \mathbb{E}_{s \in S^{\mathrm{axial}}} \underbrace{\left[ \mathbb{E}_i \left[ ||G_i^\psi(\mathbf{I}_{\mathrm{pred},*}^s) - G_i^\psi(\mathbf{I}_*^s)||_1 \right] \right]}_{\text{VGG feature covariance similarity}} \tag{5}$$

Here, the function $\psi_i$ denotes VGG-19 activation layers **relu_$i$_1** where $i \in [0,5]$, and $G_i^\psi$ is a $C_i \times C_i$ matrix constructed from $\psi_i$ with activation feature size of $C_i \times H_i \times W_i$. $\mathbb{E}_{(\mathbf{A},\mathbf{B})}$ denotes the combined mean of images $\mathbf{A}$ and $\mathbf{B}$. $\mathbf{I}^s$ is the $s^{th}$ slice of a 3D volume $I$.

## 2.4. Training and Fine-Tuning

Apdx. A details the training stages. We trained all models on an NVidia-2080 Ti GPU.
**Cross-validation training strategy.** We first trained our model using a five-fold, cross-validation strategy in the Kirby dataset. For each fold, 16 subjects were used for training and 5 for validation. Test-retest samples from the same subject were assigned to either both training or both validation to avoid leakage. For each fold, we used 210 $\mathbf{M}_{\mathrm{dr}}$ masks per model (21 subjects $\times$ 2 samples $\times$ 5 dropout masks). For data augmentation (details in Apdx. B.3), we utilized MONAI's (Cardoso et al., 2022) foreground-cropping, rotating, and flipping methods in three axes. We also used the RandCropByPosNegLabeld method to crop the original images into smaller patches of size $48 \times 48 \times 48$ voxels centered around a random voxel in $\mathbf{M}_{\mathrm{dr}}$. Loss function hyperparameters were $\lambda_1 = 1, \lambda_2 = 0.1, \lambda_3 = 0.1, \lambda_4 = 250$. We used a batch size of 4, the Adam optimizer with a momentum of 0.5, and an initial learning rate of 0.0002. After training for 80 epochs, we applied a learning rate decay strategy.
**Fine-tuning in unseen data.** We performed fine-tuning to generalize our model to the unseen VDCRA dataset. Subjects were separated into VDCRA-HC and VDCRA-MS cohorts ($n = 5$ each), and a separate network was fine-tuned for each. For each cohort, we generated 20 random $\mathbf{M}_{\mathrm{dr}}$ masks per sample. Each model was fine-tuned using a single subject and validated with the remaining four using the same augmentations as before.

| Validation Experiment | | | | | | | | L1 $\downarrow$ | PSNR $\uparrow$ | SSIM $\uparrow$ |
|---|---|---|---|---|---|---|---|---|---|---|
| | | Input Channels | | | | | | | T1w Model | |
| | | $\phi(\mathbf{I}_{T1}, \mathbf{M}_{dr})$ | $\mathbf{M}_{tis}$ | $\mathbf{I}_{tis}$ | $\phi(\mathbf{I}_{FL}, \mathbf{M}_{dr})$ | $\mathbf{E}(\phi(\mathbf{I}_{T1}, \mathbf{M}_{dr}))$ | | L1 $\downarrow$ | PSNR $\uparrow$ | SSIM $\uparrow$ |
| Input channel ablation | 3D CNN | ✓ | ✓ | | | | | 11.49 | 21.4 | 0.76 |
| | 3D CNN | ✓ | ✓ | ✓ | | | | 7.44 | 23.13 | 0.77 |
| | 3D CNN | ✓ | ✓ | ✓ | ✓ | | | 7.29 | 23.46 | 0.83 |
| | 3D CNN | ✓ | ✓ | ✓ | | ✓ | | 6.75 | 23.17 | 0.85 |
| No SPE | 3D CNN | ✓ | ✓ | ✓ | ✓ | ✓ | | 7.94 | 25.27 | 0.84 |
| Generator architectures | 2D CNN | ✓ | ✓ | ✓ | ✓ | ✓ | | 4.57 | 23.77 | 0.81 |
| | 2.5D CNN | ✓ | ✓ | ✓ | ✓ | ✓ | | 4.04 | 24.41 | 0.77 |
| | 2D LaMA | ✓ | ✓ | ✓ | ✓ | ✓ | | 3.47 | 25.67 | 0.84 |
| | 2.5D LaMA | ✓ | ✓ | ✓ | ✓ | ✓ | | **2.97** | 27.91 | 0.86 |
| Ours | 3D CNN | ✓ | ✓ | ✓ | ✓ | ✓ | | 5.37 | **31.35** | **0.86** |

Table 1: Quantitative results for validation of inpainted T1w images. This ablation study shows the contributions of the input channels, SPE as well as various generators.

## 3. Validation of Inpainted Images

Since we had access to the original images without dropout, we used them for validation of the inpainted images. We calculated the L1 error, peak signal-to-noise ratio (PSNR), and structural similarity image metric (SSIM) between the inpainted images and the original ground truth. We evaluated the contributions of the input channels, the SPE, and various generator models in ablation studies. Table 1 shows these results for one validation fold.

**Multi-modal input data.** We first investigated the effects of each input channel. We found that as each input was introduced, all three metrics improved (Table 1-pink).

**Sinusoidal positional encoding.** We found that implementing our network with SPE also improved all three metrics compared to the model without SPE (Table 1-yellow).

**Generator Architectures.** We compared our model to 2D and 2.5D generators developed for inpainting (Zhang et al., 2020; Suvorov et al., 2022). Table 1-blue shows that our model had the best PSNR and SSIM, while 2.5D LaMA (Suvorov et al., 2022) had the best L1.

**Fine-tuning.** We assessed the affect of fine-tuning in the VDCRA-HC cohort by comparing voxel intensity distributions in GM, WM, and CSF labels for the inpainted images before and after fine-tuning (Fig. 6-left). We found that performing fine-tuning led to a synthesized image histogram closer to the original image across all three tissue types, indicating a reduction in intensity variation between datasets.

## 4. Validation of the CSF Inpainting Task for Synthetic Cortical Atrophy

We next applied our model to the CSF inpainting task in images obtained using a registration-based synthetic atrophy (RBSA) method (Larson and Oguz, 2022). We compared FS cortical surfaces from the atrophied images before and after inpainting.

**Synthetic atrophy induction.** We synthetically atrophied the Kirby images using the RBSA pipeline (Larson and Oguz, 2022). We refer to the original and atrophied images as

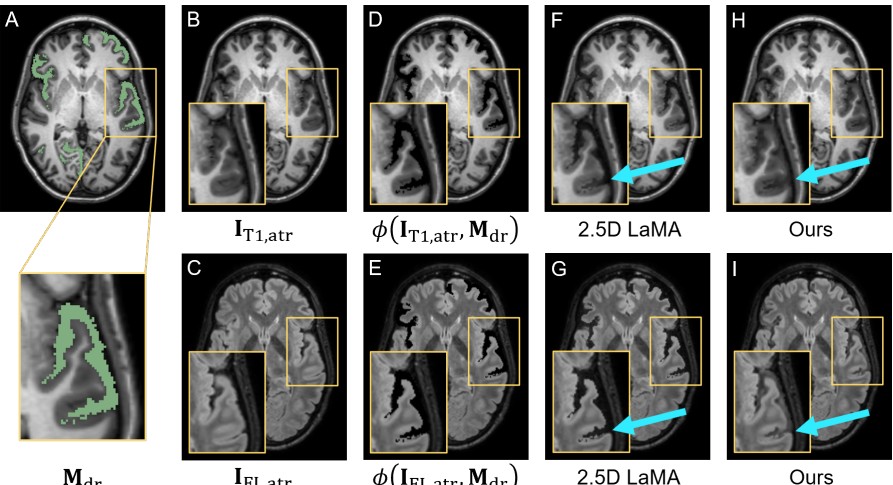

Figure 4: CSF-inpainted T1w (top) and FLAIR (bottom) images. Our model creates realistic CSF even in deep sulci, whereas 2.5D LaMA has discontinuities (arrows)

.

$\mathbf{I}_{*,\text{orig}}$ and $\mathbf{I}_{*,\text{atr}}$, where $*$ indicates T1w or FLAIR. We used the FS cortical parcellations (aparc+aseg.mgz) in one sample per subject (retest samples are not used in this experiment), and selected seven regions for atrophy induction. For VDCRA, similar to Larson et al., we created subject-specific label maps with ROIs surrounding each landmark cluster; this allowed us to also deform the landmarks. For all $\mathbf{I}_{*,\text{atr}}$, we generated tissue labellings $\mathbf{I}_{\text{tis,atr}}$ by taking the corresponding $\mathbf{I}_{\text{tis,orig}}$ and replacing the atrophied GM voxels with CSF.

**Inpainting.** Our model implementation for CSF inpainting differed slightly from training (see Apdx. A). Rather than using random patch-based dropouts, we dropped out all CSF voxels affected by atrophy induction as our $\mathbf{M}_{\text{dr}}$. This included (1) voxels classified as GM in $\mathbf{I}_{\text{tis,orig}}$ but CSF in $\mathbf{I}_{\text{tis,atr}}$, and (2) voxels surrounding the atrophied GM. We performed inpainting using the model trained on the specific dataset and modality using the following input channels: $\phi(\mathbf{I}_{\text{T1,atr}}, \mathbf{M}_{\text{dr}})$, $\phi(\mathbf{I}_{\text{FL,atr}}, \mathbf{M}_{\text{dr}})$, $\mathbf{I}_{\text{tis,atr}}$, $\mathbf{M}_{\text{dr}}$, and the edge map $\mathbf{E}(\phi(\mathbf{I}_{*,\text{atr}}, \mathbf{M}_{\text{dr}}))$.

To reduce memory consumption, we utilized a sliding window inference technique (Apdx. B.4) using overlapping patches of $96 \times 96 \times 96$ voxels, with a stride of 20 voxels between patches. Gaussian weights were applied to the patch edges to ensure smooth results.

Fig. 4 illustrates T1w and FLAIR images generated by our model alongside the non-inpainted images. As hypothesized, inpainting CSF alleviated the issue of blurring induced by RBSA methods around the GM/CSF interface, and also generated more realistic cortical thinning in narrow sulci. The 2.5D LaMA generator had discontinuities along the GM/CSF boundaries (blue arrows), despite yielding better quantitative metrics in Sec. 3. More detailed, qualitative ablation study results are available in Apdx. C.

**Cortical thickness change.** We evaluated the accuracy of CT change in the 7 atrophied regions in the Kirby dataset (Fig. 5). Error was calculated as the absolute difference between the true induced change (Larson and Oguz, 2022) and the measured change (Fischl and Dale,

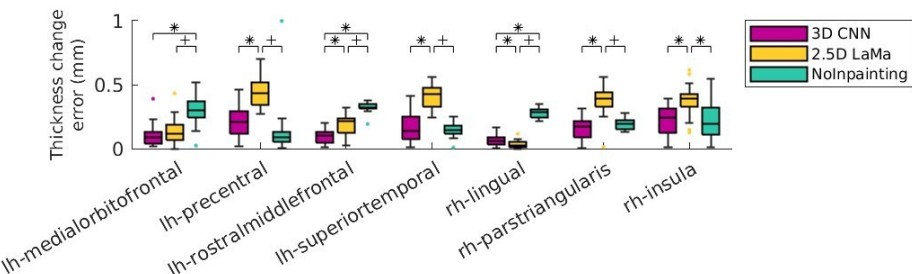

Figure 5: Mean CT change error in each atrophied region for the Kirby dataset. +: significance ($p < 0.05$), ∗: significance w/ Bonferroni correction ($n = 21$).

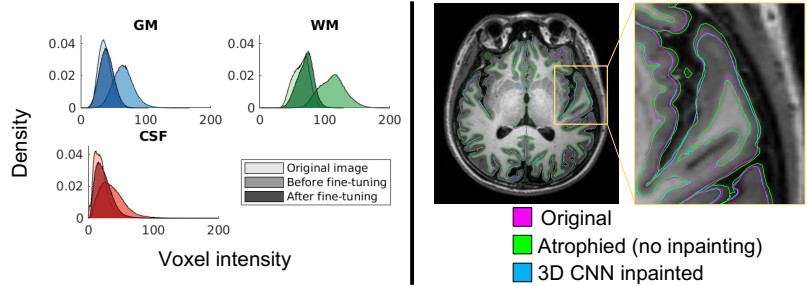

Figure 6: **Left**, GM/WM/CSF histograms in the VDCRA-HC cohort. Fine-tuning brings each histogram substantially closer to the original image. **Right**, FS surfaces overlaid on the original (no atrophy) image. Inpainting creates more realistic GM surfaces that match the atrophy (green GM surface is outside pink GM surface, but blue GM surface is correctly inside pink surface).

2000) for data inpainted using our 3D CNN, 2.5D LaMa, and without inpainting. 3D CNN inpainting either significantly improved ($p < 0.05$, paired t-tests) or had no significant effect on CT accuracy over no-inpainting. 3D CNN performed either significantly better than or comparably to 2.5D LaMA in most regions, except in the rh-lingual region where 2.5D LaMA performed significantly better. Importantly, 2.5D LaMA performed significantly worse than no inpainting in 4 out of the 7 regions, which indicates its performance is not satisfactory for the CSF inpainting task. This highlights the importance of evaluating inpainting methods in downstream tasks in addition to image quality metrics such as those in Table 1.

**Cortical surface evaluation.** We qualitatively compared the cortical surface reconstructionss for the inpainted images generated using the VDCRA dataset with fine-tuning. We found that the surfaces generated using the inpainted images better matched the induced GM atrophy. Fig. 6-right shows that FS often placed the surface for the atrophied (no inpainting) image (green) outside that of the original image (pink); this is due to blurring of the GM/CSF boundaries caused by the RBSA method.

## 5. Discussion and Conclusion

We presented a self-supervised, 3D GAN-based inpainting method that can realistically recreate multiple different tissue types in an MRI. We implemented 3D versions of several existing 2D methods including a patch-based dropout framework, edge detection for improved anatomical priors, and sinusoidal positional encoding. We showed that with fine-tuning, we can successfully apply our network to unseen data. Finally, we showed that our method improves the results of registration-based synthetic atrophy methods, and that the output can be successfully used for accuracy validation of cortical surface analyses. In future work, we will explore the performance of our model to create other types of synthetic atrophy, such as ventricular expansion and WM atrophy.

## Acknowledgments

This work was supported by the Advanced Computing Center for Research and Education (ACCRE) at Vanderbilt University.

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

## Appendix A. Overall Pipeline: Training, Fine-tuning, and Application to CSF Inpainting

Our training pipeline consists of three distinct stages, as summarized in Fig. A.1.

In the first stage, we train a generative model using a self-supervised approach. The model is capable of generating a variety of tissue types, given a dropout mask. We do this in a 5-fold cross-validation setting.

In the second stage, when dealing with an unseen dataset, we un-freeze the weights of the first phase and fine-tune the model on the new dataset. This is done by randomly sampling from the dataset and generating multiple different dropout masks.

In the final stage, using either of these models, the weights of the model are frozen and the model is applied to atrophied data with a cerebrospinal fluid (CSF) dropout, as described in Section 4. Here, we use the GM atrophy algorithm proposed in (Larson and Oguz, 2021) to induce GM atrophy in the synthetic tissue images.

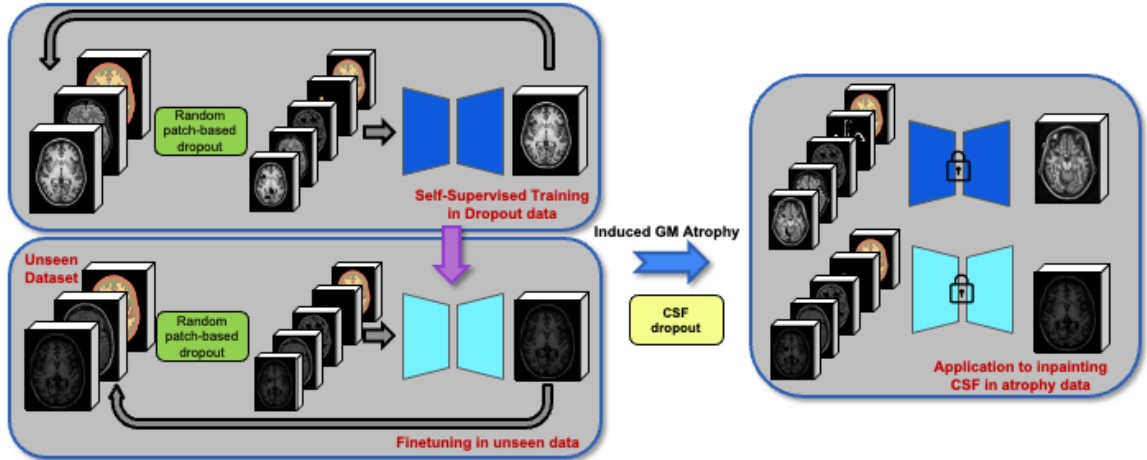

Figure A.1: Our training workflow consists of 3 main stages. In the first stage, we train a self-supervised model using random dropouts. In the second stage, we perform fine-tuning to unseen datasets. In the third stage, we freeze the model weights and apply to the CSF inpainting task.

## Appendix B. Implementation Details

### B.1. Random Patch-Based Dropout Implementation Details

For each target input image $\mathbf{I}_*$, we generated a dropout mask $\mathbf{M}_{dr}$ of dimensions $H \times W \times D$ (the same as $\mathbf{I}_*$). Let the volume of the image be $V = H \times W \times D$. We generated random dropout patches with a ratio $r$, where $r$ was randomly drawn from a uniform distribution $r \sim U(0, 0.01)$. We generated random patches until the combined volume of all patches exceeded $r \times V$. The length of the patches along each dimension was constrained to between 5% and 10% of the image size $\{H, W, D\}$, i.e.,

$$5\% \times \{H, W, D\} \leq \{h, w, d\} \leq 10\% \times \{H, W, D\}, \tag{6}$$

where $h, w, d$ denote the patch dimensions. The patch dimensions $h, w, d$ were determined by the starting position $p_{start}^k$ and ending position $p_{end}^k$ at each axis, each following a uniform distribution. Overlapping patches were accepted, but patches containing background voxels were rejected. During the self-supervised phase, we train our model to predict the dropped-out regions of the original image from the input random noise, using the original image itself as 'ground truth' for supervision. A step-by-step description of this is provided in Algorithm 1.

---

**Algorithm 1** Patch-based Dropout algorithm

**Input:** The image $\mathbf{I}$ with dimensions $H \times W \times D$

1  Max_Drop_Volume = $H \times W \times D \times \mathcal{U}(0, 0.01)$
2  **for** $K \in \{H, W, D\}$ **do**
3  $\quad$ Max_Patch_k = INT($K \times 0.1$)
4  $\quad$ Min_Patch_k = INT($K \times 0.05$)
$\quad$**end**
5  Total_Drop_Volume = 0
6  $\mathbf{M}_{\mathrm{dr}} = \varnothing(size = (H, W, D))$
7  **while** *Total_Drop_Volume < Max_Drop_Volume* **do**
8  $\quad$ **for** $K \in \{H, W, D\}$ **do**
9  $\quad\quad$ $p_{start}^k \in \mathcal{U}(0, K - Min\_Patch\_k)$
10 $\quad\quad$ $p_{end}^k = \min\{p_{start}^k + \mathcal{U}(Min\_Patch\_k, Max\_Patch\_k), K\}$
$\quad$**end**
11 $\quad$ Patch = $\mathbf{I}\,[p_{start}^h : p_{end}^h, p_{start}^w : p_{end}^w, p_{start}^d : p_{end}^d]$
12 $\quad$ **if** $0 \in Patch$ **then**
13 $\quad\quad$ **CONTINUE**
$\quad$**end**
14 $\quad$ $\mathbf{M}_{\mathrm{dr}} = \mathbb{1}[\mathbf{M}_{\mathrm{dr}} + \mathbb{1}(Patch)]$ $\triangleright$ force $\mathbf{M}_{\mathrm{dr}}$ to binary image, in case of overlapping patches
15 $\quad$ Total_Drop_Volume += volume (Patch)
**end**

---

## B.2. Network Architecture

### B.2.1. Generator

Our generator structure is shown in Fig. A.2. We use 5 different configurations of blocks, each represented by a different color. The basic building block of our generator is a 3D convolutional layer, followed by Spectral Normalization, Instance Normalization, and ReLU activation, as proposed in (Johnson et al., 2016). The kernel size of the convolutional layers varies for each block, with the blue block utilizing a kernel size of $7 \times 7 \times 7$ with stride 1, and the orange block utilizing a kernel size of $4 \times 4 \times 4$ with stride 2 for downsampling. Additionally, we include 8 residual blocks of the first layer, each utilizing a $3 \times 3 \times 3$ convolutional layer with dilation of 2. The yellow block is the transposed version of the

orange block, used for upsampling. Our generator output is produced via an activation layer, shown in teal.

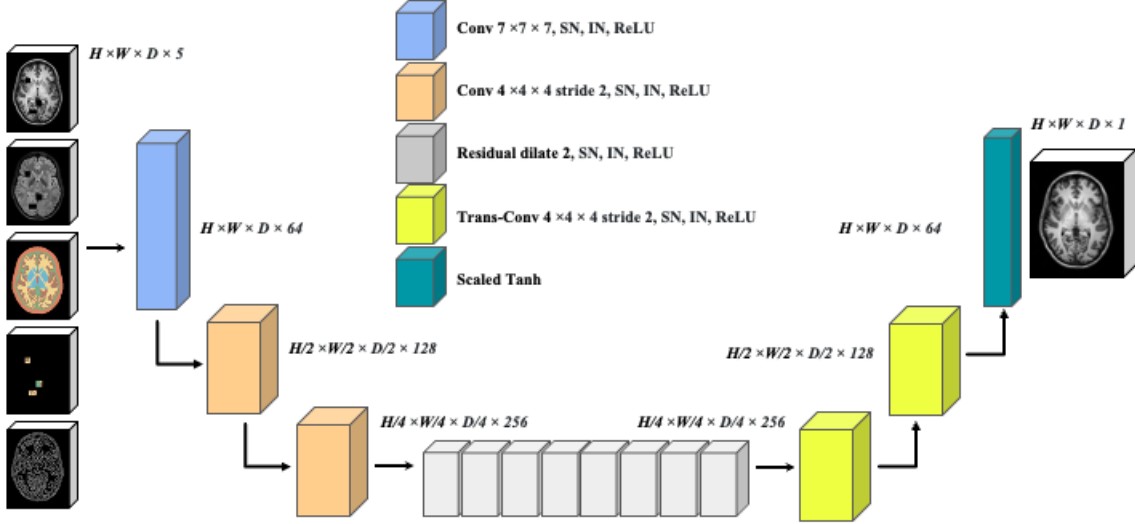

Figure A.2: Generator structure.

### B.2.2. Discriminator

The architecture of our proposed discriminator is shown in Fig. A.3 and is based on the $70 \times 70$ PatchGAN (Zhu et al., 2017). The discriminator consists of two distinct block configurations. The violet blocks are comprised of a 4x4x4 convolutional layer, followed by Spectral Normalization and LeakyReLU activation, with a stride of 2. The navy blocks, on the other hand, are also comprised of a $4 \times 4 \times 4$ convolutional layer, Spectral Normalization and LeakyReLU activation, but with a stride of 1. Each block is stacked on top of each other to form the overall discriminator architecture.

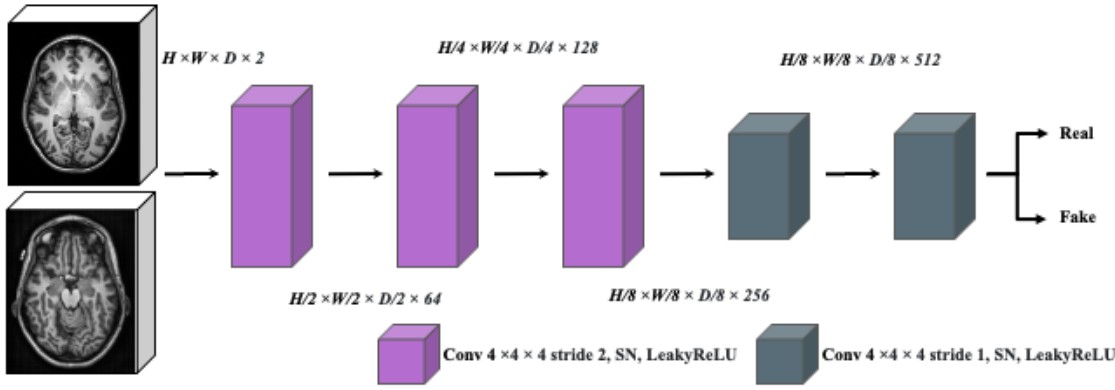

Figure A.3: Discriminator structure.

## B.3. Data Augmentation Strategy

In our proposed model, we use various data augmentation techniques during the training process to increase the robustness of our model. The specific configurations for these augmentations are listed in Table A.1, which details the parameters and probabilities used. It is important to note that these augmentations are excluded during the validation phase. We apply the intensity normalization first by mapping the intensity range $[0, 255]$ to $[0, 1]$. We also include Foreground Crop, Random Crop, Random Flip, and Random Rotate 90, and we followed the implementation of these augmentations provided in MONAI library (Cardoso et al., 2022).

| Transform | Parameters | p |
|---|---|---|
| Intensity Normalization | [0-1] | 1 |
| Foreground Crop | N/A | 1 |
| Random Crop By Pos/Neg Label | dim=(48,48,48), #=4,neg=0 | 1 |
| Random Flip | axis 0 | 0.1 |
| Random Flip | axis 1 | 0.1 |
| Random Flip | axis 2 | 0.1 |
| Random Rotate 90 | $k = 3$ | 0.1 |

Table A.1: Data augmentation during our training process. **p** indicates the probability of applying an augmentation.

## B.4. Sliding Window Inference

In this work, we implement a sliding window inference technique to address the limitation of GPU memory when processing large 3D medical image data. The implementation is depicted in Fig. A.4.

Specifically, we adopt the sliding window patch approach used in MONAI (Cardoso et al., 2022), using a patch size of $96 \times 96 \times 96$ pixels. This configuration strikes a balance between GPU consumption and the field of view (FOV) of the input data. The gap between each patch is set at 20 pixels.

After generating the inpainted image of each patch, we concatenate adjacent patches by applying a Gaussian weighting to obtain the final output to avoid seam artifacts. This approach allows us to process large 3D medical images while maintaining a high level of detail in the inpainted regions.

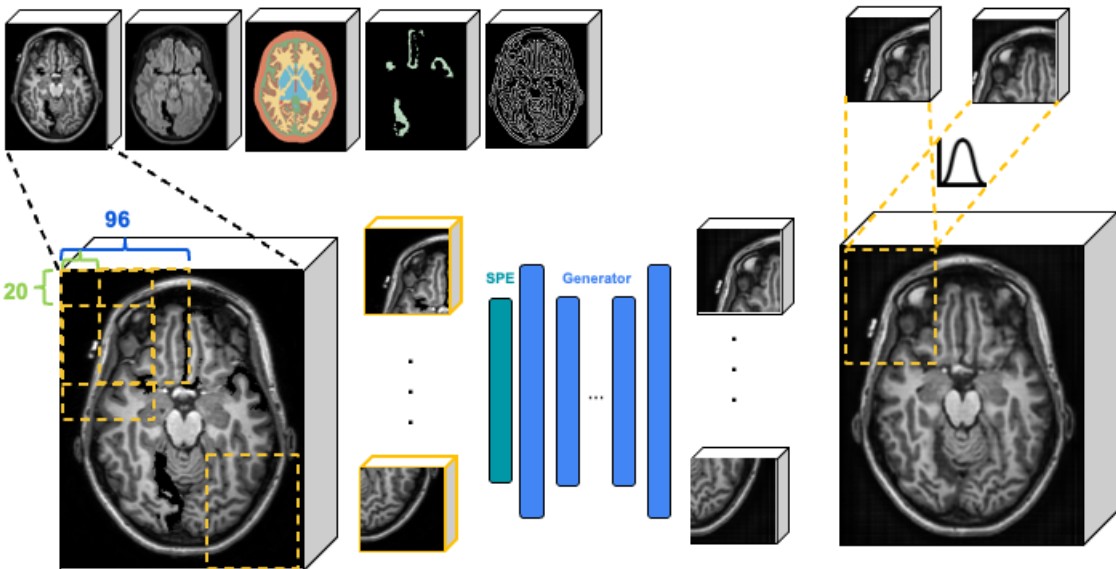

Figure A.4: An example of sliding window inference for generating a T1w image. The right top panel shows the application of Gaussian weights to generate the fused image output.

## Appendix C. Qualitative Comparison for CSF Inpainting Task

In this section, we present detailed qualitative comparison results for each model from the ablation study detailed in Sec. 3 when applied to the CSF inpainting task. Figs. A.5 and A.6 display example Kirby T1w and FLAIR image results, respectively. These images show that inpainting decreases the blurring affect induced by RBSA methods, as hypothesized. We found that although the 2D and 2.5D generators were able to produce CSF in atrophied regions, they were not able to recreate realistic atrophy, and instead resulted in jagged edges and discontinuities along tissue boundaries as well as unrealistic tissue contrast. On the other hand, our method produced an image with cleaner GM/CSF boundaries, more similar to real data. We also found that the use of SPE contributed to the synthesis of these more realistic images.

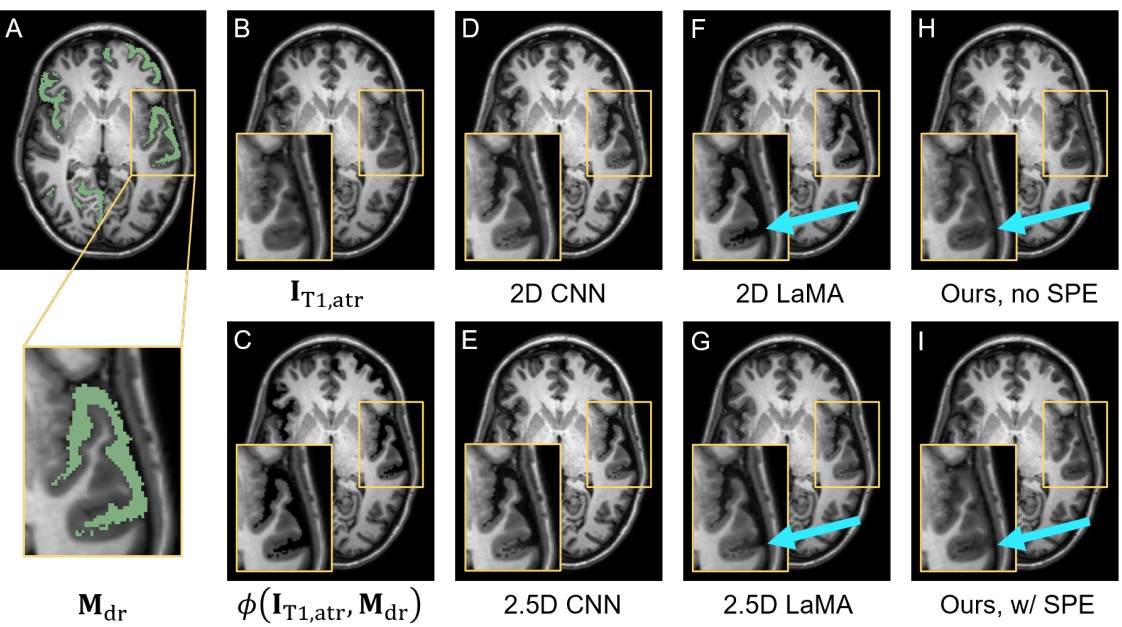

Figure A.5: T1w CSF inpainting results (Kirby). Our model synthesizes realistic images, while the other models suffer from jagged edges and discontinuities (arrows).

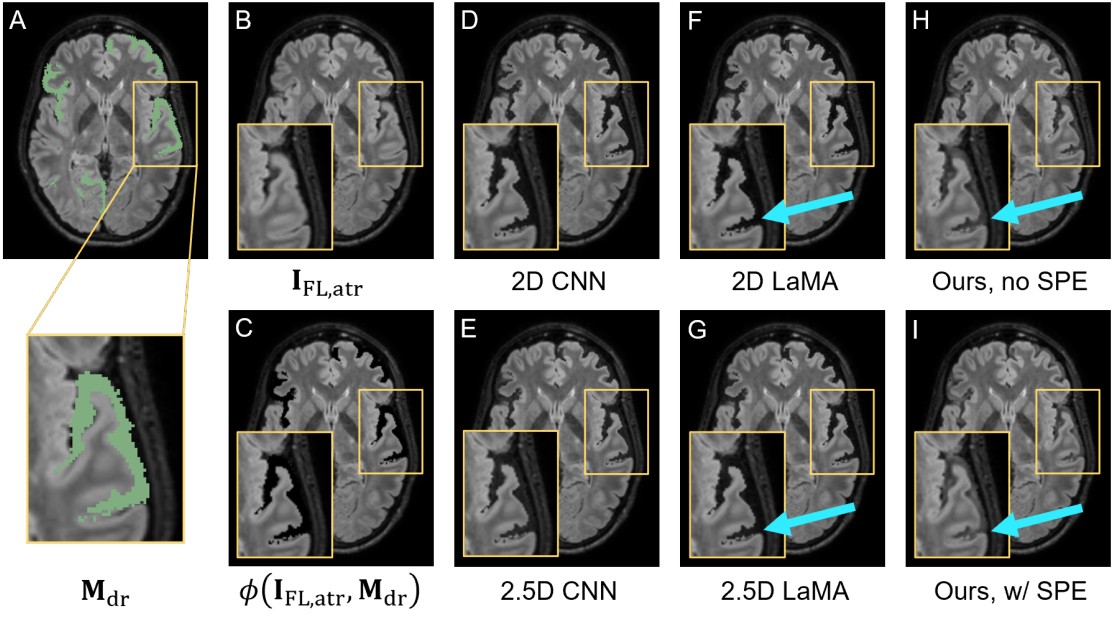

Figure A.6: FLAIR CSF inpainting results (Kirby). Our model synthesizes realistic images, while the other models suffer from jagged edges and discontinuities (arrows).

