# OpenReview forum: "Self-Supervised CSF Inpainting for Improved Accuracy Validation of Cortical Surface Analyses "
_MIDL.io/2023/Conference — MIDL 2023 Oral_

### Official Review · Reviewer_46ap · 2023-02-03

**Confidence:** 3
**Preliminary Rating:** 4
**Recommendation:** Poster

**Summary:**

The work describes a computational method to generate synthetic atrophies in neuroimaging data to validate cortical thickness estimates. To this end, the authors use a 3D GAN model that employs self-supervised learning to generate the virtual atrophied images. Combining methods previously established for 2D data (e.g. sinusoidal encoding, edge maps), the author show that their approach reduces blurring artifacts compared to registration-based reference methods.

**Strengths:**

I think the paper is well written.
The work focuses on an interesting and important application for neuroimaging research.
The results look good, although I cannot judge the numerical results as I do not have much experience with this application.


**Weaknesses:**

I do not see any major weaknesses.

Overall the paper is good. I felt some technical explanations tend to be a bit too short in the method description.
I could not follow all of the details (which is probably expected when the manuscript has to be short to adhere to the guidelines), but it might help to supply a bit more information to the reader. I would suggest revising the manuscript with this in mind and putting a bit more weight on explanations (even one sentence would help sometimes).

**Deanonymize Review:**

no

**Detailed Comments:**

In particular 2.2 generation of input image channels is a bit too condensed and might benefit from more explanations (e.g. the random patch-based dropout should be explained with a bit more detail if possible). If space is the limitation, maybe the bullet list of the papers' distribution in the introduction could be condensed/shortened.

Can you interpret the differences you find between performance in different brain regions? Apart from the statistical significance it might be interesting to get an intuition what makes a region easier or harder to inpaint.

**Paper Type:**

both

**Questions To Address In The Rebuttal:**

- As outlined above, I would suggest revising the manuscript and providing a few more explanations or motivations (but I do not consider this absolutely essential).
- Did you correct for multiple testing when stating the p-values in Fig. 5?
- If space allows I would be curious to hear the authors' thoughts on the differences in performance between brain regions (Figure 5). Do you know what characteristics made some of the regions (like lh-precentral) more challenging?

---

### Official Review · Reviewer_BGqb · 2023-02-05

**Confidence:** 4
**Preliminary Rating:** 3
**Recommendation:** Poster

**Summary:**

In this paper the authors propose an inpainting method based on a GAN to be used for the simulation of atrophy and the subsequent validation of cortical thickness measures.
They require the use of a T1 and a FLAIR image and apply their proposed framework to two datasets. In addition to qualitative measures of performance, they quantitatively assess the benefit of their method by measuring the obtained error in cortical thickness measurements in multiple brain areas and comparing it with error obtained without any inpainting.

**Strengths:**

Despite being quite niche, the motivation of the paper is clearly presented and well motivated.
The quantitative evaluation on cortical thickness measurement error does not only present numerical differences but test their significance
The method despite its complexity is relatively clearly presented and the steps are presented in a logical fashion

**Weaknesses:**

The figures sometimes do not seem to reflect the points made by the authors:
- in figure 2, the extracted patch seem to be void and not filled with Gaussian noise
- in figure 4, the picture chosen for the proposed framework appears blurrier than the one with the 2.5D LaMA solution
- the point of figure 3 is lost to the reader
In the quantitative analysis of the estimated error for cortical thickness, there is only comparison to the absence of inpainting but not with any of the other existing methods which makes the experiment suboptimal.
It is unclear why the model is to be trained on any kind of tissue when it is only and strictly applied to CSF afterwards
In this modelling it seems that only cortical atrophy is considered - there should be at least a discussion on ventricular expansion and white matter atrophy as it does not seem correct to consider all atrophy to result in cortical grey matter loss


**Deanonymize Review:**

no

**Paper Type:**

methodological development

**Questions To Address In The Rebuttal:**

Could you please particularly address the question of the atrophy model as well as the reasoning behind the two stage modelling of inpainting (generic before CSF only?)
Clarification on the figures and further indication of the performance in comparable methods would be much appreciated as well

---

### Official Review · Reviewer_feNd · 2023-02-06

**Confidence:** 3
**Preliminary Rating:** 5
**Recommendation:** Poster

**Summary:**

This paper introduces a 3D GAN technique, based on prior implementations that were (apparently) only 2D. The method is applied to generate MRI images with gray matter atrophy with improved realism attributed to dropout training, edge map priors, and sinusoidal positional encoding. The generated images are usable in downstream tasks such as cortical segmentation and thickness measurement. In my view, the fundamental contribution of this paper is the ability to in-paint 3D images using auxiliary side information.

**Strengths:**

- The writing is clear, concise and to the point
- The prior work is described well and utilized
- The presented method could be quite novel
- The work makes a lot of progress in translating many of the 2D methods to 3D
- The work has an extensive appendix with good discussion of implementation details
- The work generated images are usable on downstream tasks.

**Weaknesses:**

- The framing of this problem is somewhat unclear (I think due to brevity and space constraints). In my view, there appears to be is a gap between what is stated as the unmet need in the Introduction and what the algorithm actually does (uses auxiliary side information).
- The notation is a little hard to follow in some places, and the shorthand (e.g. for each of the loss terms) is not clear. I appreciate that this makes the paper much shorter, but some explanation (maybe under-braces in the text?) would help with clarity.
- Which norm is used in the loss functions?
- There is limited explanation of the Style and Perception Loss, specifically for the self-supervised cases. How was the 2D VGG-19 trained? (on MRI images or optical images?)
- Table 1 is clear but there is no discussion or insight into why a prior method 2.5D LaMA gets lower L1 and similar SSIM. Similarly, no explanation why the edge map but not the drop-out FLAIR image increases the SSIM in the ablation experiment?

**Deanonymize Review:**

no

**Detailed Comments:**

- Some typos
- Figure 1 can be improved for clarity, particularly in the region involving the perceptual loss. But, also, the patch-based generation is also unclear from the diagram.

**Paper Type:**

both

**Questions To Address In The Rebuttal:**

1. In my view, there appears to be is a gap between what is stated as the unmet need in the Introduction and what the algorithm actually does (uses auxiliary side information). Can you please clarify the actual problem statement? Is it: can we reconstruct T1 or FLAIR images with high accuracy and realism from T1/FLAIR images with missing patches AND complete label information AND edge information AND a dropout image M_{dr} ? I may be misunderstanding, so clarification of this point I think is essential.

2. Related to #1, please explain this sentence from the paper:
"At each iteration of adversarial training, the generator (Apdx. B.1) synthesizes the whole image. We then extract the voxels within Mdr to replace the corresponding voxels in φ(I∗,Mdr). The discriminator (Apdx. B.2) then classifies Ipred,∗ as either “real” or “fake”.  "

The operation of the network/algorithm/method is not clear. If it will help, can you explain this operation mathematically? This does not seem like a standard GAN training, even for in-painting.

3. Please explain this sentence from the paper:
"These methods can also struggle within deep sulci where CSF is not originally visible due to partial volume effects, which prevents deformable models from generating CSF in the atrophied result despite GM thinning. "

Originally visible in which image? Original T1/FLAIR images of healthy patients? of patients with real gray matter atrophy? If its not visible, how is the evaluation conducted (e.g. Figure 6) ?

4. On a related note:
"Fig. 4 displays an example of the T1w and FLAIR images generated by our model alongside the non-inpainted images. "
I do not see the non-inpainted images. Additionally, its not clear why its claimed that the presented method IS better than 2.5D LaMA. There appears to be more CSF filled in, but its unclear that the method from this paper is indeed "sharper". Explanations to this point would be helpful.

5. How is measured cortical thickness change measured on the in-painted images? Is there an automated algorithm?

---

### Meta-Review · Area_Chair_18dM · 2023-02-20

**Recommendation:** Accept (Poster)
**Confidence:** 5

**Metareview:**

**Positive:** All reviewers agree that the paper is well-written, fairly clearly presented and well-motivated. The reviewers find the paper interesting and novel.

**Negative:** The information was presented a bit too densely in some sections (given the space constraints). The authors have addressed this. Reviewer feNd has suggested some final changes for the camera-ready version, and I encourage the authors to follow this advice.

**Recommendation:** The reviews were thorough and thoughtful, and the authors used the feedback to substantially revise and improve the paper. In particular, it seems the authors have satisfactorily addressed the points raised by the most critical review. I do not hesitate to recommend acceptance.